# Identification and Characterization of an RRM-Containing, RNA Binding Protein in *Acinetobacter baumannii*

**DOI:** 10.3390/biom12070922

**Published:** 2022-06-30

**Authors:** Caterina Ciani, Anna Pérez-Ràfols, Isabelle Bonomo, Mariachiara Micaelli, Alfonso Esposito, Chiara Zucal, Romina Belli, Vito Giuseppe D’Agostino, Irene Bianconi, Vito Calderone, Linda Cerofolini, Orietta Massidda, Michael Bernard Whalen, Marco Fragai, Alessandro Provenzani

**Affiliations:** 1Department of Cellular, Computational and Integrative Biology, DeCIBIO, Proteomics and MS Core Facility, University of Trento, 38123 Trento, Italy; caterina.ciani@unitn.it (C.C.); isabelle.bonomo@unitn.it (I.B.); mariachiara.micaelli@unitn.it (M.M.); chiara.zucal@gmail.com (C.Z.); romina.belli@unitn.it (R.B.); vito.dagostino@unitn.it (V.G.D.); irene.bianconi@unitn.it (I.B.); orietta.massidda@unitn.it (O.M.); 2Magnetic Resonance Center (CERM), Department of Chemistry “Ugo Schiff”, University of Florence, 50019 Florence, Italy; rafols@giottobiotech.com (A.P.-R.); vito.calderone@unifi.it (V.C.); cerofolini@cerm.unifi.it (L.C.); 3Giotto Biotech s.r.l, Sesto Fiorentino, 50019 Florence, Italy; 4International Centre for Genetic Engineering and Biotechnology (ICGEB), Padriciano, 99, 34149 Trieste, Italy; alfonso.esposito@icgeb.org; 5Consorzio Interuniversitario Risonanze Magnetiche di Metalloproteine (CIRMMP), 50019 Florence, Italy; 6Institute of Biophysics (IBF), National Research Council (CNR), FBK Nord, 38123 Trento, Italy; mwhalen@fbk.eu

**Keywords:** *Acinetobacter baumannii*, RNA recognition motif, ELAVL1

## Abstract

*Acinetobacter baumannii* is a Gram-negative pathogen, known to acquire resistance to antibiotics used in the clinic. The RNA-binding proteome of this bacterium is poorly characterized, in particular for what concerns the proteins containing RNA Recognition Motif (RRM). Here, we browsed the *A. baumannii* proteome for homologous proteins to the human HuR(ELAVL1), an RNA binding protein containing three RRMs. We identified a unique locus that we called *AB*-*Elavl*, coding for a protein with a single RRM with an average of 34% identity to the first HuR RRM. We also widen the research to the genomes of all the bacteria, finding 227 entries in 12 bacterial phyla. Notably we observed a partial evolutionary divergence between the RNP1 and RNP2 conserved regions present in the prokaryotes in comparison to the metazoan consensus sequence. We checked the expression at the transcript and protein level, cloned the gene and expressed the recombinant protein. The X-ray and NMR structural characterization of the recombinant AB-Elavl revealed that the protein maintained the typical β_1_α_1_β_2_β_3_α_2_β_4_ and three-dimensional organization of eukaryotic RRMs. The biochemical analyses showed that, although the RNP1 and RNP2 show differences, it can bind to AU-rich regions like the human HuR, but with less specificity and lower affinity. Therefore, we identified an RRM-containing RNA-binding protein actually expressed in *A. baumannii*.

## 1. Introduction

The RNA Recognition Motif (RRM) is the most diffused RNA-binding domain present in eukaryotes and in proteins is often found in association with other domains (that could be all RRM-like or different). The single RRM domain is characterized by the presence of two consensus sequences: a highly conserved sequence (RNP1) of eight amino acids and a less conserved sequence of six (RNP2) [1,2,3]. The typical secondary structure consists of a β_1_α_1_β_2_β_3_α_2_β_4_ topology and the RNP1 and RNP2 are localized in the internal β_1_β_3_ strands [4]. The role of the eukaryotic RRM-containing protein is associated with many functions in the cell, as pre-mRNA processing, mRNA stability, translation and degradation [5]. In prokaryotes, instead, its role is still not totally elucidated, and the RRM domain-containing protein is mainly composed by a single domain of around 90 amino acids [1].

*Acinetobacter baumannii* (*A. baumannii*), is considered as one of the most concerning carbapenem-resistant bacteria, for its ability to be highly adaptable to antibiotics. *A. baumannii* is a Gram-negative opportunistic pathogen able to cause infections associated with high mortality, thanks to its exchange mobile genetic elements as transposons and plasmids [6,7].

The relevance of RBPs in *A. baumannii* is far from being elucidated, but some examples have been reported: the silencing of Csr(A), a global post-transcriptional regulator responsible for metabolisms of glucose, leads to the impairment of the growing abilities of the bacterium [8,9]; the RNA chaperone Hfq, has an important role as a virulence factor, since its knockout leads to reduced growth rate and stress tolerance [10,11]. Several RBP were described to be overexpressed in resistant strains such are enolase, RNAse E and NusA, all involved in mRNA processing and gene expression modulation [12]. However, to the best of our knowledge, the presence and role of RRM-containing RNA-binding protein in *A. baumannii* has not been reported.

One of the most studied RRM-containing RNA-Binding Protein (RBP) families among the eukaryotes is the Elav-like family (Elavl). The Elavl proteins are widely spread across all metazoans and are characterized by a high degree of conservation between different species^2^. In humans there are four paralogous genes (*ELAVL1–4*) that encode for four different proteins, HuR (or HuA), Hel-N1 (or HuB), HuC, and HuD, with different roles and cellular localization; they are constituted of three distinct RRM domains, with the most conserved sequences at the level of RNP1 and RNP2 [13]. ELAV-Like proteins bind specific sequences of RNA called AU-Rich Elements (ARE) characterized by the enriched presence of adenylates and uridylates. AREs can be located either in the intronic regions as well as in coding or non-coding parts of the mature mRNA, and contribute to mRNA splicing, maturation, stabilization, and translation [5,14,15].

Here, we identified an RRM-containing protein in *Acinetobacter baumannii* that we called AB-Elavl, starting from a protein similarity search with the human ELAV-like protein HuR. We cloned the corresponding gene, expressed the encoded protein, and characterized its biochemical function and protein structure.

## 2. Material and Methods

### 2.1. Similarity Search for Homologous Proteins to HuR Protein in A. baumannii

The human HuR protein sequence (NP_001410.2) was used as a query to search for the most similar protein in the *A. baumannii* genome, using tblastn on the NCBI web server (last accessed date 9 November 2019). The search was restricted to the species *A. baumannii* within the RefSeq Genome Database. The best scoring hit (i.e., the *A. baumannii* protein displaying highest similarity with the human HuR, F3P16_RS16475) was searched in all *A. baumannii* genomes available using tblastn, and in all other bacterial genomes using the Ortholuge database [16]. The sequences of the bacterial Elav-like proteins were submitted to the MEME-suite tool MEME v5.3.3 [17], to find conserved motifs. We performed the search using the following parameters: -mod zoops -nmotifs 50 -minw 6 -maxw 10 (that means search for at least 50 motifs occurring zero or one time per sequence and spanning 6–10 aa in length).

### 2.2. Cloning and Expression of the rAB-Elavl for Biochemical Characterization

The mRNA of the orthologous of the *A. baumannii* HuR, *AB-Elavl*, was retro-transcribed into cDNA and the sequence was amplified and inserted into the pET30a(+) vector (GenScript, Piscataway, NJ, USA) by using the forward (5′-CGGC CATATG ATACTCAAATGTATA-3′) and reverse (5′-ATAT CTCGAG CTCTTCAGCTGCCTT-3′) primers containing the NdeI and the XhoI restriction sites, respectively. Frame and sequence of the full-length ORF, with the His tag-encoding sequence located at the 3′-end, was confirmed by Sanger sequencing. The recombinant vector pET30a(+)-AB-Elavl was amplified in competent *E. coli* Top10 and the recombinant protein has been expressed into *E. coli* Rosetta BL21. Overnight cultures of *E. coli* BL21 were diluted at 1:50 with the LB medium. At OD_600_ of 0.5, cultures were induced with isopropyl β-d-thiogalactoside (IPTG) at 0.2 mM and grown overnight at 18 °C. Cells were spun down and lysed in buffer containing 20 mM HEPES pH 7.5, 300 mM NaCl, 3 mM MgCl_2_, and 0.5 mg/mL Proteases Inhibitor Cocktail (Leupeptin, Aprotinin and Pepstatin from Sigma Aldrich, St. Louis, MI, USA) and then centrifuged at 16,000× *g* for 30 min at 4 °C. The supernatant was incubated with Ni-NTA Agarose beads (Ni-NTA Agarose, Qiagen GmbH, Hilden, Germany) for 2 h at 4 °C. After washing the beads with buffer A (20 mM HEPES, pH 7.5, 150 mM NaCl, 3 mM MgCl_2_ and 20 mM imidazole), buffer B (as buffer A with 50 mM imidazole) and buffer C (as buffer A with 100 mM imidazole), protein was eluted with buffer D (as buffer A with 250 mM imidazole). The eluted protein was dialyzed against storage buffer (20 mM HEPES, pH 7.5, 150 mM NaCl, 3 mM MgCl_2_, 5% glycerol) and stored at −80 °C [18]. Recovered recombinant protein was analyzed by Coomassie staining on 12%-SDS PAGE. The relative protein concentration was determined in three different ways: using bovine serum albumin (BSA) standards and densitometry quantification (ImageJ 1.4 software, NIH) of corresponding bands on acrylamide gels, using the Bradford assay and by UV-vis spectrometry using the molar extinction coefficient.

### 2.3. Expression and Purification of rAB-Elavl for X-ray and NMR Analysis

Recombinant AB-Elavl (rAB-Elavl) protein encoded in plasmid pET-30a(+) was overexpressed in BL21(DE3) GOLD cells. Cells were grown in LB or M9 minimal media supplemented with ^15^NH_4_Cl or ^15^NH_4_Cl and ^13^C-glucose at 37 °C until optical density (OD_600_) reached 0.6–0.8. Subsequently, protein production was induced with 0.2 mM of isopropyl β-d-thiogalactoside (IPTG), cells were incubated at 18 °C overnight and harvested by centrifugation at 4 °C, for 15 min at 7500 rpm. Cell pellet was resuspended in lysis buffer (50 mM HEPES, pH 6.8, 300 mM NaCl, 3 mM MgCl_2_, Proteases Inhibitor Cocktail), ruptured by sonication and separated by centrifugation at 30,000 rpm for 35 min at 4 °C. Soluble fraction was collected and treatment with 5% PEI solution was performed to remove DNA/RNA attached to the protein. Re-suspension of the protein was performed with the lysis buffer. Soluble protein was filtered with a 0.22 µm membrane and purified by a Ni^2+^-affinity chromatography step using a His-Trap HP 5 cm^3^ column previously equilibrated in 50 mM HEPES pH 6.8, 300 mM NaCl, 3 mM MgCl_2_, Proteases Inhibitor Cocktail. r*AB*-Elavl was eluted with increasing concentration of imidazole (20–50–100–250 mM) in the buffer and subsequently dialyzed overnight against 4 dm^3^ of 20 mM HEPES buffer at pH 6.8, containing 150 mM NaCl and 3 mM MgCl_2_. The protein was filtered and further purified to homogeneity by size exclusion chromatography using a Hi load 26/60 Superdex 75 pg column that was previously equilibrated in 20 mM HEPES pH 6.8, 150 mM NaCl, 3 mM MgCl_2_ and Proteases Inhibitor Cocktail.

### 2.4. Crystallization of rAB-Elavl

rAB-Elavl was concentrated to 6 mg/mL in 20 mM HEPES buffer pH 6.8, containing 150 mM NaCl, 3 mM MgCl_2_ and Proteases Inhibitor Cocktail. Crystals diffracting at 1.6 Å were obtained by sitting drop vapor diffusion at 293 K, in which 5 µL of protein solution was mixed with 5 µL of reservoir solution and suspended over 600 µL of the same reservoir solution. The reservoir solution consisted of 0.1 M sodium acetate trihydrate pH 4.5, 3 M sodium chloride.

### 2.5. X-ray Data Collection and Refinement

The dataset was collected in-house, using a BRUKER D8 Venture diffractometer equipped with a PHOTON III detector, at 100 K (Bruker, Billerica, MA, USA); the crystals used for data collection were cryo-cooled using 25% ethylene glycol in the mother liquor. The crystals diffracted up to 1.6 Å resolution: they belong to space group I4_1_ with one molecule in the asymmetric unit, a solvent content of about 50%, and a mosaicity of 0.3°. The data were processed using the program XDS [19], reduced and scaled using XSCALE [19] and amplitudes were calculated using XDSCONV [19]. The structure was solved using the molecular replacement technique; the model used was obtained through MODELLER [20] by using 1FXL as the template. The successful orientation hand translation of the molecule within the crystallographic unit cell was determined with MOLREP [21]. The refinement was carried out using PHENIX [22], applying TLS restraints. In between the refinement cycles, the model was subjected to manual rebuilding using COOT [23]. The quality of the refined structures was assessed using the program MOLPROBITY [24]. Data collection and refinement statistics are summarized in Table 1. The relevant coordinates and structure factors have been deposited at the Protein Data Bank under the accession code 7QZP.

### 2.6. NMR Measurements and Protein Assignment

Experiments for backbone assignment were performed on samples of the ^13^C, ^15^N isotopically enriched RRM domain of rAB-Elavl at protein concentration of 300 µM in buffer solution (20 mM HEPES, pH 6.8, 150 mM NaCl, 3 mM MgCl_2_, Proteases Inhibitor Cocktail). NMR spectra were recorded at 298 K on a Bruker AVANCE NEO 900 spectrometer, equipped with a triple-resonance Cryo-Probe (Bruker, Billerica, MA, USA). Spectra were processed with the Bruker TOPSPIN software packages and analyzed with CARA (Computer Aided Resonance Assignment, ETH Zurich). The backbone resonance assignment of RRM domain was obtained by the analysis of 3D HN(CO)CA, 3D HNCA, 3D HNCO, 3D HN(CA)CO, 3D CBCA(CO)NH and 3D HNCACB spectra [25]. Secondary structure prediction was performed with TALOS+ [26] by using the chemical shifts of HN, N, C’, Cα, and Cβ as input data.

### 2.7. Titration of rAB-Elavl with RNA Probes

The effect of two different types of RNA (AREpos and AREneg) on the ^15^N-isotopically enriched RRM domain of AB-Elavl (70 µM) was evaluated in the following experimental conditions: 20 mM HEPES, pH 6.8, 150 mM NaCl, 3 mM MgCl_2_, Proteases Inhibitor Cocktail. 2D ^1^H ^15^N BEST-TROSY. NMR spectra were acquired at 298 K on Bruker Avance III and AVANCE NEO NMR spectrometers operating respectively at 950 and 900 MHz (^1^H Larmor frequency) and equipped with triple-resonance Cryo-Probes, to monitor the effect of increasing amounts (17.5, 35, 52.5, 70, 140 µM) of each RNA added to the protein solution.

### 2.8. RNA-Electrophoresis Mobility Shift Assay (REMSA)

rAB-Elavl protein (at indicated concentrations) and RNA probes with DY681 infra-red tag (at a concentration of 2.5 nM) (Eurofins Genomics, Ebersberg, Germany) were incubated in REMSA buffer (20 mM HEPES pH 7.5, 50 mM KCl, 450 µM BSA, 0.25% Glycerol) in a final volume of 20 μL at room temperature. The reaction mix was then loaded onto 6% native polyacrylamide gel containing 0.5% Glycerol. Run was performed in 0.5X TBE buffer at 80 V for 40 min and then 100 V for 20 min, at 4 °C. Free and complexed RNA probes were detected with Odyssey infrared Imaging System (LI-COR Odissey Infrared Imager Biosciences, Lincoln, NE, USA) using filters for red light emission detection [27,28,29].

### 2.9. Amplified Luminescent Proximity Homogeneous Assay (ALPHA Screen)

AlphaScreen assays have been performed using histidine (nickel) chelate detection kit (Histidine detection kit Nickel Chelate 6760619C, PerkinElmer, Waltham, MA, USA) in white 384 Optiplates. AlphaScreen assay was applied to study the interaction between rELAV-like protein and the different biotinylated single-stranded probes: ARE pos (5′-Bi-AUUAUUUAUUAUUUAUUUAUUAUUUA-3′), ARE pos 19 (5′-Bi-AUUAUUUAUUAUUUAUUUA-3′), ARE pos 11 (5′-Bi-AUUAUUUAUUA-3′) and ARE neg (5′-Bi-ACCACCCACCACCCACCCACCACCCA-3′) (Eurofins Genomics, Ebersberg, Germany). All reagents were reacted in ALPHA buffer (25 mM HEPES pH 7.4, 100 mM NaCl, 0.1% BSA). For the optimization of the assay, the optimal protein:RNA ratio (hook point) was identified: a series of concentrations of the recombinant protein (0–40 µM) were incubated with different concentrations of ARE pos probe (0–500 nM). For the EC_50_ calculation 500 nM of the rAB-Elavl protein was incubated with a series of concentrations of probes (0–500 nM) for 15 min at room temperature, then anti-His-Acceptor beads (20 μg/mL final concentration) and Streptavidin-Donor beads (20 μg/mL final concentration) were added, and the reaction was incubated in the dark at room temperature for 60 min to reach equilibrium. Fluorescence signals were detected on Enspire plate reader instrument (PerkinElmer; 2300 Multilabel Reader, PerkinElmer, Waltham, MA, USA). Non-specific interference with the assay has been evaluated by reacting the same amount of acceptor and donor beads (20 μg/mL/well) without the probe and with just the protein buffer in the same experimental conditions. The half maximal effective concentration (EC_50_) was calculated with GraphPad Prism software v6.1 [27,28,29].

### 2.10. Time Course Experiments Kinetic

Time course experiments were carried out incubating in a final volume of 20 μL, a series of concentrations (0–50 nM) of the RNA probes (Bi-AREpos and Bi-AREneg) with a constant concentration of rAB-Elavl protein (500 nM), anti-His-Acceptor beads (20 μg/mL) and Streptavidin-Donor beads (20 μg/mL) in Alpha buffer. Assays were performed in triplicate. The wells were all seeded with a cocktail containing Alpha buffer and beads, while rAB-Elavl protein and probes were added in a second moment, according to the time checkpoints. The signals of the whole 384-well plate were detected at the end of the time course. Association and dissociation rate constants were determined from nonlinear regression fits of the data according to the association kinetic model of multiple ligand concentration in GraphPad Prism^®^, version 6.1. The resulting K_D_ values obtained by koff/kon ratio [29].

### 2.11. Western Blot from A. baumannii and HEK293 Cells Lysate

*A. baumannii* strain ATCC 19606 (American Type Culture Collection, Manassas, VA, USA), was grown in Luria-Bertani (LB) medium, in the incubator at 37 °C, 200 rpm shaking. Inoculum was grown overnight and the next day it was diluted to a final concentration of 0.05 OD_600_. The bacteria were allowed to grow to a final OD_600_ of 0.5 (they were measured at the spectrophotometer at a λ: 600 nm) and pelleted at 4000 rpm for 20 min at 4 °C. The pellet was incubated for 30 min in ice with lysis buffer (50 mM tris HCl pH7.5, 100 mM NaCl, 10% glycerol, 0.1% triton, 1 mM DTT, 1 mM EDTA, Leupetine, Aprotinin, Lysozyme, 2.5 U/μL) to a final volume equal to 1/20 of the initial culture, and then sonicated. HEK293 cells were rinsed with PBS and lysed in ice-cold RIPA buffer, while bacteria were lysed in a bacterial lysis buffer (20 mM Tris HCl pH 8, 150 mM KCl, 1 mM MgCl_2_, 1 mM DTT, DNAse, Proteinase inhibitors and RNAse inhibitors). Proteins were boiled in SDS gel sample buffer, separated by SDS-PAGE and immunoblotted onto a polyvinylidene difluoride membrane. The primary antibody against AB-Elavl was developed by Davids biotechnologie in rabbit, while the antibody against HuR was purchased from Santa Cruz Biotechnologies (6A97) (Santa Cruz, Dallas, TX, USA). Bands were visualized with anti-rabbit or anti-mouse HRP-conjugated secondary antibodies and scanned on Biorad Chemidoc (Biorad, Hercules, CA, USA).

### 2.12. Time Resolved Fluorescence Resonance Energy Transfer (HTRF-FRET)

All assays were performed in 20 μL in 96-well low-volume white plates, in triplicate. The EC_50_ calculation was performed by adding increasing concentrations of RNA. The experiments were performed by incubating the protein with the RNA for few min before to add the mix composed of beads (Acceptors beads europium-labeled anti-6X His-Antibody and donor beads XL665–conjugated for biotin detection at a final concentration of 35 nM), potassium fluoride buffer and FRET reaction buffer 1x provided by the manufacturer. After brief spinning (1000 rpm, 1 min), the plate was incubated for 1 h at 4 °C. The signals of acceptors and donors were detected using Tecan Spark (Tecan, Zürich, Switzerland) and the results were calculated using the following equation:Acceptors/Donors × 10,000.

### 2.13. Immunoprecipitation (IP) Assay of AB-Elavl

For each IP, 2.5 mg of total protein lysate from *A. baumannii* was used. Bacteria were lysed in RIP lysis buffer (20 mM Tris HCl pH 8, 150 mM KCl, 1 mM MgCl_2_, 1 mM DTT, DNAse, Proteinase inhibitors and RNAses inhibitors) [30]. The lysate was incubated with Pierce A/G beads (Thermo Scientific Pierce 88847–88848, Waltham, MA, USA) for pre-clearing steps 2 h at 4 °C; in parallel, 50% A and 50% G beads were incubated either with 10 μg of anti-rAB-Elavl antibody or 10 μg of IgG antibodies for antibody-coating step for 2 h at RT. At the end of the 2 h of incubation, the protein lysate was incubated with antibodies and beads overnight at 4 °C. Finally, samples were washed (5 times, 5 min each wash) with NT2 buffer (50 mM Tris-HCl pH 7.4, 150 mM NaCl, 1 mM MgCl_2_, 0.05% NP40). The pellet was then analyzed by western blot or mass spectrometry assay.

### 2.14. Mass Spectrometry (MS) Analysis

To perform the MS analyses of rAB-Elavl protein and *A. baumannii* ATCC 19606 lysate, the samples were separately resolved on 10% polyacrylamide gel. After Coomassie stain, a protein band corresponding to rAB-Elavl and a region representing proteins with molecular masses of 6–14 kDa were cut from the gel. Excised gel bands were cut into small pieces (~1 mm^3^) and subjected to reduction and alkylation with 10 mM DTT and 55 mM iodoacetamide, respectively. Gel pieces were then dehydrated with acetonitrile and dried in a speed-vac. Gel plugs were rehydrated with 50 mM NH_4_HCO_3_ solution containing 12.5 ng/mL trypsin (Promega, Madison, WI, USA) on ice for 30 min. The digestion was continued at 37 °C overnight. The supernatant was collected, and the peptides were sequentially extracted from the gels with 30% ACN/3% TFA and 100% ACN. All of the supernatants were combined and dried in a SpeedVac. The peptides were then acidified with 1% TFA, desalted on C18 stage-tips and resuspended in 20 μL of 0.1% formic acid buffer for LC-MS/MS analysis.

To perform the IP-MS analyses, the co-immunoprecipitated complexes were eluted with Laemmli buffer containing 5% β-mercaptoethanol at 80 °C for 10 min. The samples were loaded on a 10% SDS-PAGE and run for about 1 cm. Gels were then stained with Coomassie and the entire stained area was excised as one sample. The stained bands were then subjected to in gel digestion and peptide desalting process as described above. Samples Digested peptides were separated using an Easy-nLC 1200 system (Thermo Scientific, San Jose, CA, USA) on a reversed-phase column (25 cm column, inner diameter of 75 µm, packed in-house with ReproSil-Pur C18-AQ material: 3 µm particle size, Dr. Maisch, GmbH), heated at 40 °C, with a two-component mobile phase system of 0.1% formic acid in water (buffer A) and 0.1% formic acid in acetonitrile (B). The 85-min gradient was set as follows: from 5% to 25% over 52 min, from 25% to 40% over 8 min and from 40% to 98% over 10 min at a flow rate of 400 nL/min. Peptides were analyzed in a Fusion Tribrid mass spectrometer (Thermo Fisher Scientific) in data-dependent mode and positive mode (2100 V). The full-scan in the Orbitrap was performed at 120.000 fwhm resolving power (at 200 m/z) and followed by a set of (higher-energy collision dissociation) MS/MS scans over 3 s cycle time. The full scans were performed with in a mass range of 350–1100 m/z, a target value of 1 × 106 ions and a maximum injection time of 50 ms. A dynamic exclusion filter was set at 40 sec. The MS/MS scans were performed at a collision energy of 30%, 150 ms of maximum injection time (ion trap) and a target of 5 × 10^3^ ions. Peptides searches were performed in Proteome Discoverer software version 2.2 (Thermo Fisher Scientific) against the *A. baumannii* database (uniprot, downloaded March 2021), the rAB-Elavl amino acid sequence, and a database containing common contaminants. Proteins were identified using MASCOT search engine, with a mass tolerance of 10 ppm for precursor 0.6 Da for product. Trypsin/P was chosen as the enzyme with 5 missed cleavages. Static modification of carbamidomethyl (C) with variable modification of oxidation (M) and acetylation (protein N-term) were incorporated in the search. The false discovery rate was set to 1% at both peptide and protein level. The results were filters to exclude potential contaminants. For protein quantification in IP–MS experiment, peak intensities were transformed into log_2_ space. Data were normalized by the average of its abundance within each sample to account for variation in sampling volumes [31]. Significant abundance differences between conditions were determined using a *t*-test.

## 3. Results

### 3.1. Identification of a Putative RRM Containing RBP, AB-Elavl

We performed a similarity search for the HuR protein in the RefSeq Genome Database, limited to the species *A. baumannii*, using tblastn and we found 25 hits on the same gene locus from different *A. baumannii* genomes (Appendix A). However, in 11 cases, the region of homology was limited to the RRM1 (aa 20–98 on the human protein), in the remaining 14 hits instead covered the RRM3 (aa 244–322, Figure 1A). The percentage of identity ranged 32.90–35.48% (on average 34.08 ± 1.00%) when the subject sequence was the human RRM1, and it ranged 31.94–46.00% (on average 35.70 ± 3.67%) when the subject was the human RRM3 (Figure 1B). The positive matches ranged 50.67–60.26% when the subject was RRM1 (on average 54.11 ± 3.15%), and 51.39–60.00% when the subject was RRM3 (on average 53.84 ± 2.66%) (Appendix A). The identified gene, F3P16_RS16475 (Appendix A), encodes for a putative RNA binding protein that has been named here AB-Elavl. The similarity scores between the human HuR’s RRMs and the AB-Elavl protein are in the same range as the similarity scores among the three RRMs (RRM1-RRM2: 32% id and 57% positives, RRM1-RRM3: 36% id and 53% positives, RRM2-RRM3: 30% id and 48% positives). The gene locus *AB-Elavl* was present in nearly all the deposited *A. baumannii* genomes (4946 out of 4972 available), suggesting that it belongs to the core genome of this species. The *AB-Elavl* gene was comprised between a gene encoding an ASCH domain containing protein, 8 bp downstream, and a gene encoding ATP-dependent helicase, 74 bp upstream (Figure 1C). We searched for AB-Elavl homologous proteins in all bacterial genomes contained in OrtholugeDB^17^ and found 227 hits in genomes spanning 12 bacterial phyla. The sequence length ranged 78–241 aa with an average of 106.68 ± 30.06; multiple sequence alignment consisted of a 375 aa alignment. The bacterial proteins homologous to the human ELAV-like family shared an identity score ranging from 9.1% (*Dyadobacter fermentans* versus *Shewanella pealeana*)–99.3% (between two *Shewanella* spp.), having on average 31.1 ± 10.8% (Appendix A). The visual evaluation of the multiple-sequence alignment suggested that there were conserved regions within the bacterial homologous of HuR (Figure 1D), so we ran the web tool MEME for motif discovery. We found that there were two motifs which were significantly conserved across all sequences (Figure 1E), one had the pattern (I/L)(Y/F/L)YGNL (*p*-value 3.0e^−1314^), the second (K/R)GF(G/A)FVEM (*p*-value 3.0e^−1407^). Those two patterns match the locations and the order of the ribonucleoprotein motifs RNP-1 and RNP-2 in each of the RRMs in HuR protein^2^, further supporting its potential RNA binding ability. Collectively, we identified a conserved gene in the *A. baumannii* species and in many other bacteria phyla, containing a RRM domain with a different consensus sequence with respect to the metazoan one.

### 3.2. AB-Elavl Gene Is Expressed and Translated in A. baumannii

We checked whether the RNA transcript corresponding to *A. baumannii* Elav-like (*AB-Elavl*) was expressed. We retro-transcribed the total RNA of the *A. baumannii* reference strain (ATCC 19606) and amplified by PCR the surroundings of the gene of interest (Figure 2A) using three different pairs of primers. The amplicons’ sequences were confirmed by Sanger sequence analysis and matched the DNA deposited sequence. We observed that our gene of interest is expressed and contained into a longer, polycistronic, mRNA of at least 764 bp (Appendix A). We cloned the *AB-Elavl* sequence into the expression plasmid pET30a(+) in frame with a 6XHis tag in the C terminal region and expressed the recombinant *A. baumannii* Elav-like (rAB-Elavl) protein in *E. coli* Rosetta BL21. We purified the recombinant protein (predicted MW 12.8 KDa) from cell lysate by affinity purification using Nickel NTA agarose beads. The purity of the rAB-Elavl protein was evaluated by Coomassie staining of protein polyacrylamide gel of each purification step performed by increasing imidazole concentration (Figure 2B). The yield of protein expression was 12.5 mg/L. The purified rAB-Elavl protein was subjected to mass spectrometry analysis after trypsin digestion. We obtained 80.2% coverage of the entire recombinant sequence, missing the first 17 amino acids, and the detected peptides perfectly matched the predicted amino acid sequence (Figure 2C up). To evaluate whether the polycistronic mRNA is translated into a protein containing the domain of interest, we performed proteome analyses by mass spectrometry on the protein lysate of *A. baumannii* ATCC 19606. Protein cell lysate was separated into a polyacrylamide gel, a band (6–14 kDa) comprising the MW of the predicted protein (predicted MW 10.8 kDa) was cut, trypsin digested and submitted to LC-MS/MS analysis. Among the detected fragments, we obtained 35% coverage of the recombinant protein with complete matching of the experimental amino acid sequences with the reference (Figure 2C down). Interestingly, the detected protein fragments contained the region of the highly conserved octapeptide KGFGFVEM that we found conserved in the protozoans and that corresponds to the ribonucleoprotein motif 1 (RNP-1) in metazoans (Figure 2D). This analysis confirmed the presence of several small peptides belonging to our hypothesized protein but did not provide any information on the real length of the translated protein. To obtain more insight on the presence and on the MW of the hypothetical AB-Elavl protein translated in the bacterium, we developed an antibody against the recombinant protein (αAB-Elavl). Rabbits were immunized with the denatured rAB-Elavl protein, and after two cycles of immunization, the serum was collected, and the IgG titer quantified. Specificity of the IgG in recognizing the protein of interest was investigated by performing western blot against the rAB-Elavl protein, the recombinant HuR (rHuR), human cell lysate (MCF7 cell line) and the total proteome of *A. baumannii* (Figure 2E). We confirmed that the immunized serum recognizes the rAB-Elavl, but not HuR, and observed a band at a slightly heavier MW compared to the calculated one in the total protein lysate of the bacterium. This suggests that the native protein is longer than the predicted one as well as the recombinant one is digested in any part during the protein production into *E. coli*. To obtain more proofs about the presence of AB-Elavl protein, we performed a protein immunoprecipitation from the lysate of *A. baumannii* using the immunized serum with the αAB-Elavl. Protein precipitate was run on polyacrylamide gel, but no bands were detected (Figure 3A). Therefore, we performed an immunoprecipitation followed by mass spectrometry (IP–MS) on the same material, to investigate which proteins were enriched, with respect to rabbit IgG, used as control. About 5675 proteins (Figure 3B) were enriched into the immunoprecipitated material and, among the most enriched ones, we found three entries in the Uniprot database related to hypothetical RNA-binding proteins of *A. baumannii* (Figure 3C and Appendix A) that are extremely similar to our protein of interest (percentage of homology 54.5–81%, Figure 3C). The three entries are D0CAL6, 86 aa, predicted MW 9560.22 Da, A0A009GG82, 79 aa, predicted MW 8715.23 Da and A0A4R5S8D9, 58 aa, predicted MW 6445.52 Da. All of them showed a predicted MW lower than the recombinant protein. Notably, in addition to the previously identified protein fragments in the protein lysate, we found eight more amino acids that completed the retrieval of the hexapeptide conserved sequence (ILVRNL) in the RNP-2 protozoa sequence. Taken together, these data indicate that the DNA encoding the hypothetical AB-Elavl protein is effectively transcribed and translated into a protein that contains the two RNPs amino acid sequences responsible for nucleic acid binding.

### 3.3. AB-Elavl Protein Has a Typical Single RRM Domain Structure

The X-ray structure (Table 1) showed the domain of an RRM domain: β1-α1-β2-β3-α2-β4 [32] with an additional β5 segment present right before the C-terminus of the protein. The two conserved amino acid sequences, ILVNRL and KGFGFVE, are localized at the level of the two internal strands of the β-sheet: β2-β3 (Figure 4A and Appendix A). The structure was solved using the molecular replacement method; the model showing the highest sequence identity (about 40%) was 1FXL (HuD, HuR human paralog in complex with AU rich element of the C-FOS RNA). MODELLER was used based on this structure to generate the model with the correct sequence for molecular replacement. The solved structure shows the absence of the first 18 residues in the electron density with respect to the cloned sequence. It is not trivial to tell whether this is due to their high mobility or rather to their loss due to some protein degradation before/during crystallization. Figure 4A shows the superposition between rAB-Elavl (red) and 1FLX (green). It appears quite clear that the fold of rAB-Elavl is very similar to that of the model used for structure solution and, in turn, similar to the typical RRM (Figure 4B). The greatest discrepancy between the two structures is in the region involving residues from 50 to 58, just before RNP-1. In our case electron density is missing for those residues, confirming thus very high mobility. The average B-factor along the sequence is, in fact, about 35 Å^2^, confirming an overall rigidity of the structure, with the only exception being the above-mentioned region 50–58 where temperature factor values are extremely high. This mobility is not present in the case 1FXL because this region interacts with RNA. The Ramachandran plot is good for all residues except for those in the mobile regions, where the chain tracing is very approximate.

The 2D ^1^H–^15^N HSQC spectrum of the r*AB*-Elavl shows well-dispersed and resolved signals in agreement with a small, uniform and well-folded protein structure (Figure 5). The backbone assignment of the protein was obtained from the analysis of triple resonance spectra. All the residues from Lys-22 to Glu-101 were assigned in the spectra, while the first 21 N-terminal residues were missing. The NMR assignment of AB-Elavl (residues Lys22-Glu101) has been deposited in the BMRB database under the accession code: 51440. According to TALOS+ predictions, the RRM domain is constituted by two α-helices and four β-strands (Appendix A), in agreement with the currently resolved crystal structure and with the reported structures of the RRM domains.

### 3.4. AB-Elavl Binds AU Rich RNA Probes

Since AB-Elavl is an RRM domain containing protein characterized by the two conserved RNPs, we investigated whether the RRM domain of AB-Elavl protein could have RNA-binding abilities. We investigated whether in the proteome of *A. baumannii* there are proteins able to recognize and bind the ARE sequences taken by the 3′UTR of *TNFα*, a target of HuR, using non-denaturing and non-cross-linked RNA ElectroMobility Shift (REMSA) assay^28^. The single strand (ss) RNA probe ARE probe was bound with an infra-red tag DY681 (IR-ARE pos). By mixing higher concentrations of the protein lysate with a fixed 2.5 nM concentration of the IR-ARE pos probe, we observed a decreased quantity of free RNA probe and the formation of a protein–RNA complex. This indicates the presence of one or more proteins able to bind the IR-ARE pos (Figure 6A). We then evaluated if the rAB-Elavl was able to bind to probes that contained the HuR consensus sequence (ARE pos) and probes without the HuR consensus sequence (ARE negative RNA probes, ARE neg), by REMSA. We mixed increasing amounts (40 nM, 80 nM and 160 nM) of protein with 2.5 nM IR-ARE pos and 2.5 nM IR-ARE neg. As shown in the mobility shift assay, rAB-Elavl clearly caused the RNA probe electrophoretic retardation detectable as one prominent band, with both probes, showing a binding preference towards the ARE pos probe in this biochemical condition, since the shifted band is not clearly visible for the ARE neg (but the free RNA decreases with the incrementation of the protein) (Figure 6B). We evaluated the possible formation of a super-shift by adding the antibody against rAB-Elavl. We were expecting the formation of the heavy complex antibody–protein–RNA, but unfortunately, we could not observe any super shift (Figure 6C).

The interactions of the AB-Elavl with ARE pos and ARE neg were also investigated through solution NMR. In the presence of ARE pos, at the protein/RNA molar ratio of 1:2 a global decrease of signal intensity was observed (Figure 7 and Appendix A). In particular, some residues located on the β-platform (Leu24, Val25, Asn27, Ser31, Val52, Thr58, Gly63, Gly65, Phe66, Lys80, Lys88, Gly89, Ile92) experience a larger decrease in signal intensity (Figure 7A and Figure 7C top). Some of these residues (or the neighboring ones) are also affected by Chemical Shift Perturbation, CSP (Val25, Arg30, Ser31, Val52, Gly63, Gly89, Ile92, Glu99, Leu100, Glu101, Figure 7B and Figure 7C top). It should be noted that Leu24, Val25 and Asn27 are located in RPN1, and Gly63, Gly65, Phe66 in RPN2. The superimposition of the X-ray structures (Appendix A) shows that the same conserved protein regions are involved in the interaction of the HuR human paralog with the AU-rich element of the C-FOS RNA. In the presence of ARE neg, instead, the effect at the same protein/RNA molar ratio is much reduced. However, some residues of the β-platform still experience a decrease in signal intensity (Val25, Leu38, Phe41, Val51, Val53, Phe64, Gly65, Phe66, Tyr90, Glu101; Figure 7A and Figure 7C bottom) and/or a CSP (Lys22, Val25, Arg30, Phe64, Gly65, Ile92, Glu98, Glu99, Leu100, Glu101 Figure 7B and Figure 7C bottom).

### 3.5. AB-Elavl Binds AU Rich RNA Probes with Nanomolar Affinity

To quantitatively characterize the binding activity of the rAB-Elavl to different RNA probes, we applied AlphaScreen technology using 5′-biotinylated ssRNA probes as substrate. We used the 5′-biotinylated ARE pos (Bi-ARE pos) and the biotinylated ARE negative (Bi-ARE neg). We optimized the assay to identify the best molar ratio between the two interacting partners coupled with anti-His-Acceptor and Streptavidin-Donor beads; the optimal concentration, before the hooking effect, was observed at 250 μM and 50 nM for rAB-Elavl and Bi-probes, respectively (Appendix A). We then evaluated the affinity of binding between the rAB-Elavl protein and probes with different sequences but the same length. The recombinant protein shows a high affinity for Bi-ARE Pos (EC_50_ = 0.5 nM) while it has low affinity for the Bi-ARE neg probe (EC_50_ = 257.1 nM) (Figure 8A,B). Coherently with REMSA, rAB-Elavl bound both probes, but we could quantify a stronger affinity for the ARE-positive probe. We then evaluated the minimal ARE sequence required for binding. We observed that rAB-Elavl, as its human orthologous^14^, has a higher affinity for longer ARE sequences than for shorter ones (EC_50_ ARE pos 29 nt = 35.62, ARE pos 19 nt = 64.76, ARE pos 11 nt = not converged) (Figure 8C). The affinity evaluation was confirmed using the HTRF–FRET assay. The probes used were the same of the AlphaScreen assay: Bi-ARE pos and Bi-ARE neg. This assay, as well, was optimized to identify the best molar ratio between the two interacting partners coupled with europium-labeled anti-6X His-Antibody and XL665–conjugated for biotin detection; the optimal protein concentration, before the hooking effect, was observed at around 200 nM for both the probes (Appendix A). The EC_50_ (nM) are respectively: ARE pos EC_50_ = 35.11 nM and ARE neg- EC_50_ = 945.5, (Figure 8D). To further define the binding affinity between the r*AB*-Elavl protein and ARE sequence, we performed a time course experiment in which different concentrations of the Bi-ARE pos were mixed to the protein with different incubation time. The experiment shows that the binding of r*AB*-Elavl to Bi-AREpos probe was both time and dose dependent (Figure 8E). Data were globally fitted using the association kinetic model of multiple ligand concentration: derived association (kon of 2.035 M^−1^ min^−1^) and dissociation (koff of 0.02687 min^−1^) rates indicated a very high affinity of the rAB-Elavl protein towards this RNA substrate and a low dissociation rate. According to the law of mass action, the equilibrium binding constant K_D_ calculated as koff/kon was obtained as K_D_ value of 13.2 nM. We performed the same type of assay for Bi-ARE neg, for which the binding resulted as ambiguous (Figure 8F).

## 4. Discussion

We uncovered the existence of an RRM-containing, RNA-binding protein in *A. baumannii*, solved its structure and provided an initial characterization of its RNA-binding abilities. We started by performing a similarity search using the HuR protein sequence as seed and found a conserved gene locus encoding for a putative RBP conserved in most of the *A. baumannii* species. This supports the hypothesis that *AB-Elavl* found in the *A. baumannii* genome encodes for a protein that has a functional relevance in *A. baumannii* physiology. The length of the AB-Elavl roughly corresponds to a single RRM, while in Eukarya, the RRM domain is present in tandem with other heterologous or homologous domains. Indeed, evolution led to an increase in the number and specificity of eukaryotic RBPs; they are often characterized by a repetition of domains that collaborate for a better affinity to the target RNA [5,14,33]. On the contrary, bacteria tend to be more streamlined, with simpler RBPs composed by just one single domain but with wider functions, since they are less specific for their targets [4,8]. Bacterial RBPs, contrary to eukaryotic ones, can normally bind a wider number of sequences. For example, Hfq has a wide substrate selection, underling the different roles covered by this protein: from RNA chaperone to ribosome biogenesis, DNA compaction, protein–protein interactions, and involvement in RNA degradation machinery [8,34,35]. We characterized the crystal structure of rAB-Elavl, which retraces the common structure of the classical RRM domain: β1-α1-β2-β3-α2-β4, in which RNP1 and RNP2 are in the internal strands of the beta-sheet (β1-β3) [1,4]. We observed a partial divergency in the RNP1 and RNP2 sequences, the regions responsible of the binding with the RNA [1,2], between the HuR’s RRMs and the AB-Elavl RRM. Hence, the target consensus sequences between bacterial AB-Elavl and eukaryotic Elavl RNPs, are characterized by similar recognition motifs [2]; however, the two proteins do not show completely equivalent RNA binding properties. Indeed, we tested the binding affinities of RNA probes using the sequences known to be targeted by HuR (ARE pos) and the respective negative sequence (ARE neg). The presence in the proteome of *A. baumannii* of possible proteins binding the ARE sequences was confirmed by a REMSA assay, and the ability of rAB-Elavl to interact both with the ARE-positive and ARE-negative probes by AlphaScreen, FRET and NMR. All the assays showed a preference of the protein toward the AU-rich sequence in the nanomolar range, even though the AC-rich peptides are also bound but with a clear lower affinity. By a time course kinetic assay, we also calculated a K_D_ in the nanomolar range (13.2 nM), although an order of magnitude higher than the reported HuR K_D,_ in the same experimental condition (2.5 nM) [29]. The kinetic experiment using the ARE neg probe, instead, gave an ambiguous K_D_ calculation.

## 5. Conclusions

All these results suggest that *A. baumannii* express an RRM-containing RBP that shares RNA-binding properties and characteristics with the human HuR for the ability to bind RNA AU-rich region, although with lower affinity and specificity. However, the exact length of the protein produced has not been determined and it is likely that the RRM domain is contained into a longer protein. The structure of the bacterial RRM domain appears very similar to the eukaryotic one but for the presence of an additional short β strand and a more flexible central region. Notably the amino acids present in the RNP1 and RNP2 are different between protozoans and metazoans, but they are similarly involved in the RNA binding. In addition, functional studies are needed to understand the role of this protein in the bacteria.

## Figures and Tables

**Figure 1 biomolecules-12-00922-f001:**
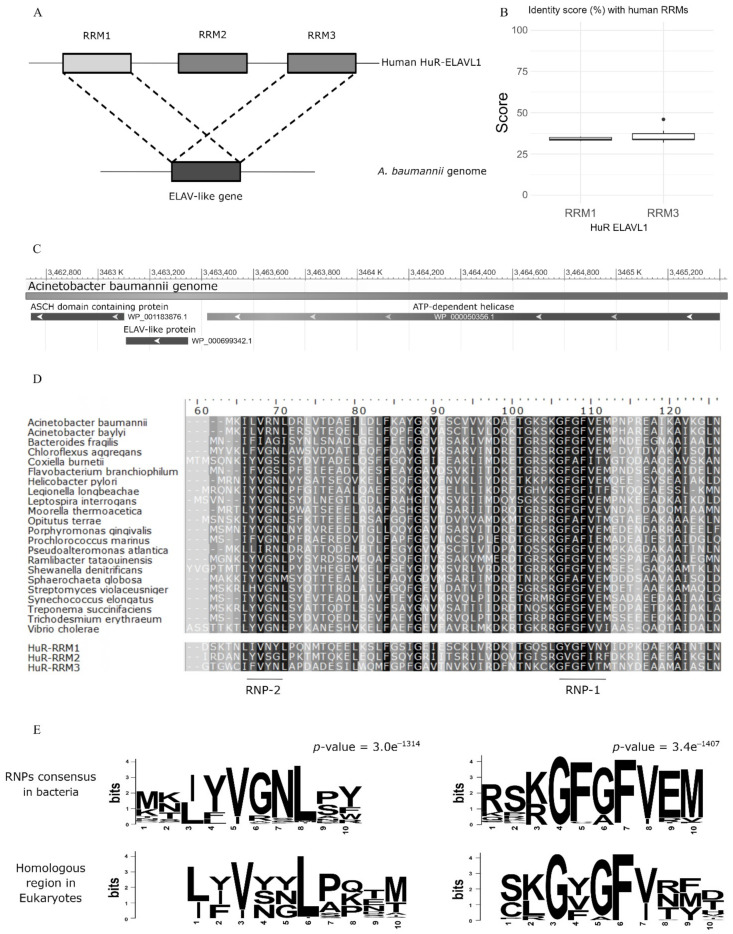
In silico analysis. (**A**) The tblastn search, using as query the human HuR and restricting the search to *A. baumannii* genomes, gave 25 hits corresponding to the same orthologous protein which share a high homology with both RRM1 and RRM3. (**B**) Boxplots showing the percentages of identity with the two RRMs, see Appendix A tblastn for the extended dataset. (**C**) Genomic context of the bacterial HuR, it is shown that the three genes, namely an ATP-dependent helicase, the AB-Elavl and the ASCH domain containing protein are arranged in proximity. (**D**) Alignment of the bacterial homologues of human ELAV in selected bacterial species with clinical or environmental relevance spanning seven phyla, along with the HuR RRMs. The background shades denote the level of conservation in that position, darker background mean more conserved residue in that position. (**E**) Sequence logos for the significantly conserved regions, corresponding to RNP1 and 2 in Prokaryotes (upper row, dataset produced in this study) and Eukaryotes (lower row, dataset from Samson 2008). The seqlogos have been aligned to highlight the presence of conserved residues.

**Figure 2 biomolecules-12-00922-f002:**
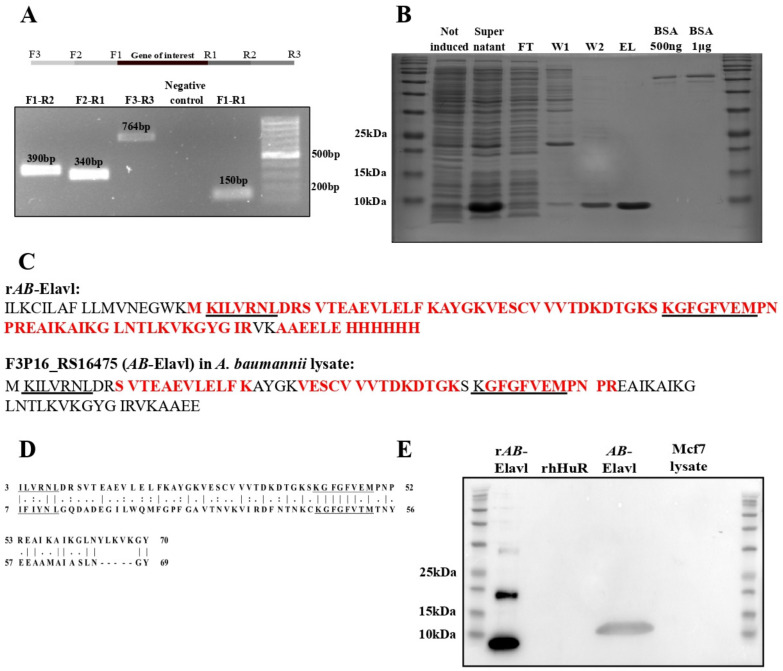
Protein identification and purification. (**A**) PCR amplification of the transcription of the polycistronic mRNA containing the sequence of interest. The amplicons produced are 390 bp for F1-R2, 340 bp for F2-R1, 764 bp for F3-R3 and 150 bp for F1-R1 (this amplicon was also used as positive control). Neg_ctrl: negative control. (**B**) Purification of the recombinant protein. FT: flow through, W: wash, EL: elution. (**C**) Mass spectrometry analysis. The recombinant protein was analyzed at first to confirm the sequence. It was then used as a reference for the analysis of *A. baumannii* proteome. In red: peptides retrieved with high confidence, underlined: conserved peptides. The predicted molecular weight is 12 KDa for the recombinant protein and 10.8 for the protein from *A. baumannii*. The predicted isoelectric point is 9.06 for both the proteins. (**D**) Alignment of *AB*-Elavl (above) and the RRM3 domain of HuR (below). “|” means that the residues in column are identical.; “:” means that the amino acid in column has been substituted by one with similar characteristics; “.” means that semi-conserved substitutions are observed. (**E**) Western blot analysis to confirm the presence of the protein of interest in the protein lysate of *A. baumannii* and in the MCF7 lysate, as well as on the recombinant proteins *AB*-Elavl and HuR.

**Figure 3 biomolecules-12-00922-f003:**
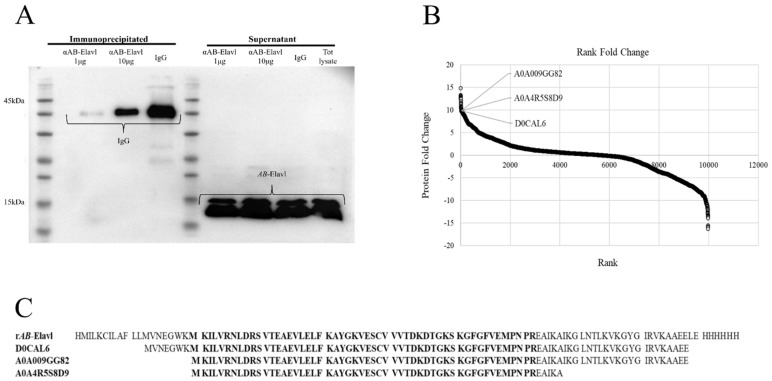
Molecular characterization of AB-Elavl. (**A**) Immunoprecipitation assay on the total protein lysate of A. baumannii. IgG was used as a control. No enrichment of the protein was visible by western blot analysis. (**B**) Protein ranking based on log_2_ fold-changes (IP/IgG) for all the proteins identified by MS showing an enrichment of three hypothetical and highly similar RNA-binding proteins (RBPs) in the top ten proteins. (**C**) Entry code and amino acid sequence of the three hypothetical RBPs based on the IP-MS analysis compared with the rAB-Elavl sequence. Bold retrieved peptides (sequence coverage: 62%, 67% and 91% for D0CAL6, A0A009GG82 and A0A4R5S8D9 proteins, respectively).

**Figure 4 biomolecules-12-00922-f004:**
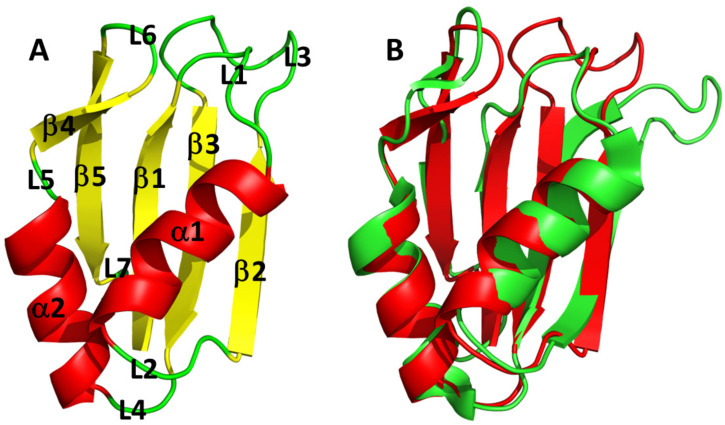
AB-Elavl protein structure. (**A**) Ribbon representation of the three-dimensional structure of the bacterial hypothetical HuR RRM domain. The secondary structural elements and loops have been annotated: helices (α1–α2), strands (β1–β5), loops (L1–L7). (**B**) Superposition of the crystal structure of the bacterial hypothetical HuR RRM domain (rAB-Elavl, red) and 1FLX (green).

**Figure 5 biomolecules-12-00922-f005:**
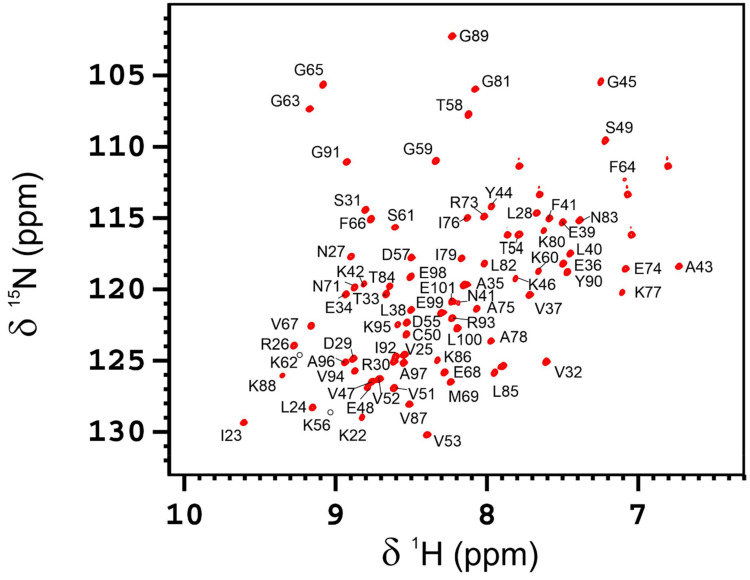
2D ^1^H-^15^N HSQC spectrum of AB-Elavl. The spectrum was recorded with a spectrometer operating at 900 MHz and 298 K. Assignment is reported on the signals.

**Figure 6 biomolecules-12-00922-f006:**
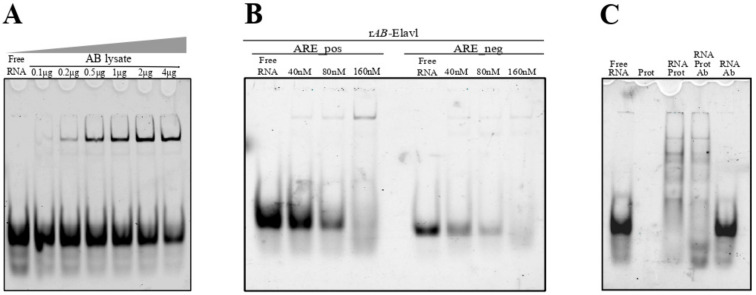
Analysis of the protein binding abilities. (**A**) REMSA assay on the total protein lysate of *A. baumannii* and a probe mimicking the AU rich sequence of TNFα (ARE pos) with an infrared tag. (**B**) REMSA assay on the recombinant protein incubated with different probes with an infrared tag: ARE pos and ARE neg. ARE pos is bound with a high affinity, while ARE neg shows a lower affinity. (**C**) REMSA assay for detection of the super-shift in presence of the antibody against *AB*-Elavl. The super-shift is not detectable.

**Figure 7 biomolecules-12-00922-f007:**
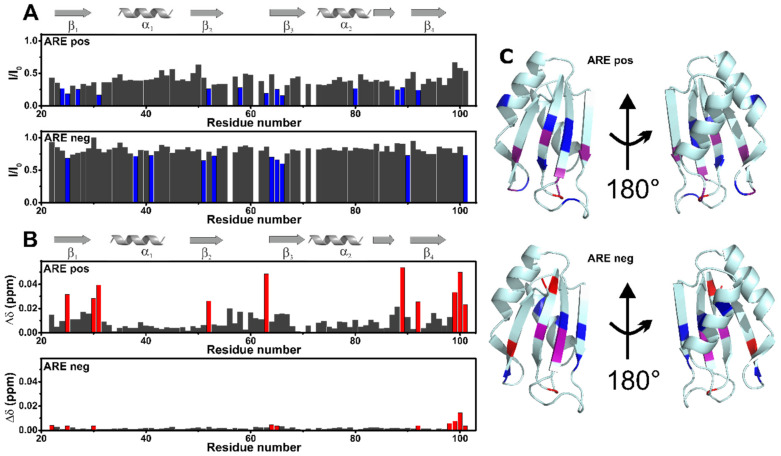
NMR analysis of the protein binding abilities. (**A**). Plots of decreases in signal intensity of rAB-Elavl RRM domain in the presence of 140 µM ARE pos (top), or 140 µM ARE neg (bottom) with respect to the free protein (70 µM). The residues experiencing the largest decreases have been highlighted in blue. (**B**). Chemical shift perturbation (CSP) of r*AB*-Elavl RRM domain (70 µM) with respect to the protein in the presence of 140 µM positive RNA (top), and 140 µM negative RNA (bottom). The CSP was evaluated with the formula: $∆\delta =\frac{1}{2}\sqrt{∆\delta_{H}^{2}+{\left(∆\delta_{N}/5\right)}^{2}}$. The residues experiencing the largest CSP have been highlighted in red. (**C**) Highlighted in blue are the residues experiencing the largest decreases in signal intensity, in red the residues experiencing the largest CSP, and in violet the residues experiencing the largest decreases in signal intensity and CSP, in the presence of 140 µM positive RNA (top), and 140 µM negative RNA (bottom).

**Figure 8 biomolecules-12-00922-f008:**
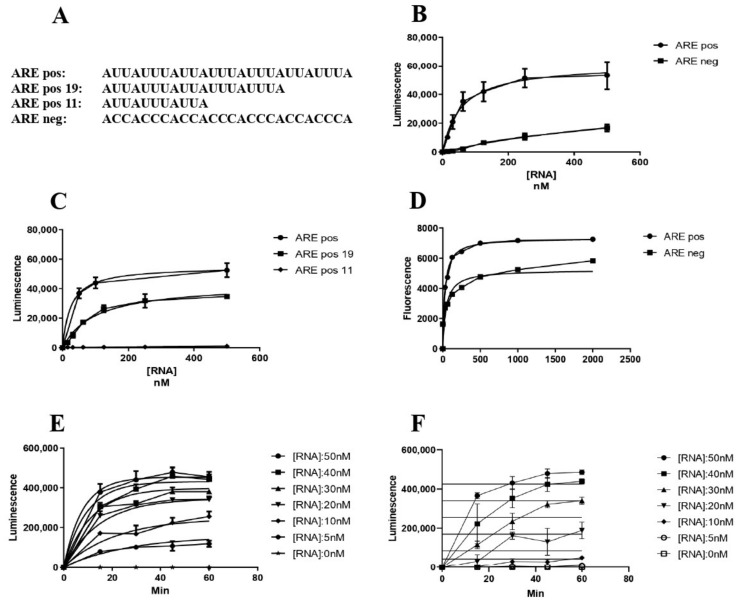
Biochemical characterization of the protein binding ability. (**A**) Sequences of the probes used in the different assays. (**B**) AlphaScreen saturation experiment between the recombinant protein and AREpos and AREneg. The EC_50_ was determined from non-linear regression fits of the data according to the dose–response model in GraphPad Prism^®^, version 6.1. (**C**) AlphaScreen saturation assay for detection of the minimal probe length for binding of the protein. The probe are AREpos with 3′ deletions: ARE pos: ARE sequence full length, ARE pos 19: ARE sequence with 19 nucleotides, ARE pos 11: ARE sequence with 11 nucleotides. The minimal number of nucleotides in order to obtain the binding is 19, but longer sequences have a higher affinity. The EC_50_ was determined from nonlinear regression fits of the data according to the dose–response model in GraphPad Prism^®^, version 6.1. (**D**) EC_50_ evaluation through saturation experiment by HTRF-FRET and AREpos and AREneg. AREpos is confirmed to have high affinity while AREneg is not well bound. The EC_50_ was determined from nonlinear regression fits of the data according to the dose–response model in GraphPad Prism^®^, version 6.1. (**E**,**F**) Kinetic experiment with rAB-Elavl and AREpos (**E**) is dose dependent, while for AREneg (**F**) the binding resulted ambiguous. Association (Kon) and dissociation (Koff) rate constants were determined from nonlinear regression fits of the data according to association kinetic model of multiple ligand concentration in GraphPad Prism^®^, version 6.1.

**Table 1 biomolecules-12-00922-t001:** Xray structure parameter of rAB-ELAV, Pdb code: 7QZP. Statistics for the highest-resolution shell are shown in parentheses.

Parameter	Values	Parameter	Values
Wavelength (Å)	1.541	R-free	0.2280 (0.3590)
Resolution range	17.39–1.654 (1.713 –1.654)	CC (work)	0.953 (0.523)
Space group	I 4_1_	CC (free)	0.935 (0.489)
Unit cell (Å)	69.56	Number of non-hydrogen atoms	642
	69.56		
	32.46		
Total reflections	96620 (2480)	Protein	601
Unique reflections	8226 (502)	Solvent	41
Multiplicity	11.7 (4.9)	Protein residues	78
Completeness (%)	87.01 (53.41)	RMSD (bonds) (Å)	0.013
Mean I/sigma(I)	18.74 (1.52)	RMSD (angles) (°)	1.74
Wilson B-factor	24.38	Ramachandran favored (%)	96.05
R-merge	0.07861 (0.8945)	Ramachandran allowed (%)	1.32
R-meas	0.08199 (0.9341)	Ramachandran outliers (%)	2.63
R-pim	0.02281 (0.4356)	Rotamer outliers (%)	6.15
CC1/2	0.999 (0.364)	Clashscore	12.87
		Average B-factor (Å^2^)	36.18
Reflections used in refinement	8214 (501)	protein (Å^2^)	36.45
Reflections used for R-free	411 (25)	solvent (Å^2^)	32.25
R-work	0.2049 (0.3354)		

## Data Availability

The coordinates and structure factors obtained by X-ray crystallography have been deposited at the Protein Data Bank under the accession code 7QZP. The NMR assignment of rAB-Elavl (residues Lys22-Glu101), generated during the current study, is available in the BMRB database under the accession code: 51440.

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
