# Peer review of "Identification and Characterization of an RRM-Containing, RNA Binding Protein in Acinetobacter baumannii"

_biomolecules, 2022, doi:10.3390/biom12070922_

Round 1
Reviewer 1 Report
In this study, Ciani and coworkers perform a similarity search using an RRM containing protein, human HuR, to identify an A. Baumannii protein that contains a single RRM, and
name it as AB-Elavl. The authors further demonstrate that this AB-Elavl protein is actually present in the bacterial, solve its crystal structure, and test its RNA binding properties.
Overall, the study is technically sound and generally well described, I have 2 major criticism and several minor criticisms. Assuming that the authors can address these criticisms, I believe that this study would represent some contribution to the field.
Major points
1. In section 3.4, the authors claim that AB-Elavl binds AU rich RNA probes, but in the Figure 6B and 6C, we do not see significant mobility shift. Even in figure 6A. when the lysate amount is doubled and doubled, we cannot see the increased intensity for the lower migrating band. These data concerns me since the ABElavl-RNA complex band mystically disappears. I would suggest use a different method to confirm the RNA binding property, for example “Isothermal titration calorimetry” or other methods.
2. In Figure 8D, seems like the ARE pos and ARE neg have comparable binding signals, the author should address this or repeat the experiment. Figure 8E and
8F are really hard to read, also add the plot showing how the nano molar binding affinity is calculated.
Minor points
1. Page 1, line 8, my understanding is HuR and ELAVL1 is the same protein,
different name, to avoid confusion, the authors should change the text to
“HuR(ELAVL1), an RNA…”
2. Page 1, line 38, add a reference here for the RRM containing protein functions.
3. Page 3, line 141, which PDB model was used for the molecular replacement should be noted here.
4. In supplementary table 1. The authors should add the PDB code in this table, that would be easier for reader.
5. Page 9, line 356, “the yield of protein expression was 11.8 uM (125.5ng/uL)”. uM is concentration unit, not yield. It means nothing about yield without the volume. The authors should change the unit to mg, or ng of protein. Also, indicate how many liters of E. coli that is starting with.
6. Page 9. The sentence across line 380-line 382. First, the post-translational modification in bacterial is very rare. Second., I don’t understand why the native protein can be longer than what the authors identified, so, the author indicates the sequence they identified is not right?. Last. if the protein is degraded during prep, we should be able to see the result from mass spec, if so, the authors should discuss with result from figure 2C,upper. Also ,do we see the N terminal residues missing in the NMR experiment? An easy way to test if it will degrade or not during prep, is making a poly His tag on the N termini of this protein.
7. Supplementary Table 3. label the residue numbers in the protein sequence
8. Figure 2C, the author should add the predicted protein sequence here as a reference, easier for readers to compare. Also “the bold” residues seem not easy to see, maybe use some colors, or a different font to distinguish them.
9. Figure 2D, the resolution of this graph is too low.
10. Figure 3A, add the MWs for the standard marker.
11. Page 11, Line 422, this protein has a beta 5 in figure 4B. is it still typical RRM?
12. Page 11, Line 422, we don’t usually call 6 protein residues “hexamer”, or 8 protein residues “heptamer”.
13. Figure 4A, it would be better to align the structures with the presence of RNA , since PDB 1FXL has RNA ligand in its model. I want to suggest trying to find the residues that are essential for RNA binding by this alignment, perform mutation on those residues, and carry the RNA binding assays with the mutant proteins to see if diminished affinities are observed. This should strengthen this manuscript significantly, but might take a lot more work. I will let the editor to decide if it is recommended or not.
14. Page 15, line 504, it should be section 3.5, not 3.4
Reviewer 2 Report
The manuscript ‘Identification and characterization of an RRM-containing, RNA binding protein in Acinetobacter baumannii’ by Caterina Ciani et al. describes characterization of an RNA binding protein from a gram-negative bacteria Acinetobacter baumannii. Authors have done phenomenally good work in the current manuscript focusing on RNA binding protein research using bioinformatics tools, structural biology methods, and biochemical characterization. Though the manuscript is written very well, however, it can be improved by adding a separate heading of Conclusion with more thoughts in the concluding remarks.
The specific comments, which could help to improve the manuscript, are:
- In the materials method section:
line 72: accession number (Uniprot or Genbank) of query and target sequence can be added in appropriate places.
line 91,110: either A600 or OD600 any one format should be used throughout the manuscript.
line 95: 16,000xg is more suitable than 16.000xg.
line 97: Hepes or HEPES, I would prefer to write HEPES throughout.
line 114,120,126, 223, 242, etc: ‘Inhibitor protease’, ‘protease inhibitor’, ‘proteinase inhibitors’, make sure these terminologies are throughout the manuscript should be consistent. Are you sure the Protease inhibitors in each buffer used is the same as discussed in line 94? We don’t expect the Size exclusion buffer should contain a protease inhibitor cocktail.
- In the X-ray Data collection and refinement section:
line 142: The search model used for MR was obtained from MODELLER. However, I would emphasize to include PDBs used by Modeller software for template design.
line 147: Supplementary Table 1 is an important result table and must be included in the main text of the article.
line 174: 0.5 ug BSA doesn’t look correct. It should be a concentration of either uM or %.
line 220: Lysozime (replace with Lysozyme)
line 216: OD should be replaced with OD600.
3. In the results section: I don’t see any supplementary tables attached with manuscript (either these have not been provided by Editors or uploaded by the authors).
line 335: replace Hur with HuR.
Figure4: 7QZP the structure determined by the authors should be mentioned clearly and presented First (A) and then comparison with 1FLX (B) While superposing crystal structure with 1FLX, keep the same orientation in both figures to understand better. Crystal structure should also highlight important findings RNA binding surface, conserved residues, etc. Also, there is no comparison of crystal structure vs NMR structure. The crystal structure is poorly described, would be better to re-write in detail.
Figure 6: REMSA or EMSA make it consistent with the text
Figure 2B: What is EL?
Line 603, 612: Why two times author contributions?
